# Evidence-Based Practice Competency of Registered Nurses in the Greek National Health Service

Stefania Schetaki, Evridiki Patelarou, Konstantinos Giakoumidakis, Christos Kleisiaris and Athina Patelarou *

Department of Nursing, School of Health Sciences, Hellenic Mediterranean University, 71410 Heraklion, Greece; ddk33@edu.hmu.gr (S.S.); epatelarou@hmu.gr (E.P.); kongiakoumidakis@hmu.gr (K.G.); kleisiaris@hmu.gr (C.K.)
* Correspondence: apatelarou@hmu.gr

**Abstract:** Nurses' competency toward evidence-based practice (EBP) has been extensively investigated by several studies worldwide. However, factors affecting the competence of Greek nurses working in the NHS have not been fully investigated in terms of EBP. Thus, this study aimed to explore the impact of the individual qualifications of nurses on their competence toward EBP. Data from 473 registered nurses working in 10 hospitals in the Greek National Health Service (NHS) were collected between October and December 2020 using a convenience sampling method in a cross-section design. The Greek version of the 35-item (five-point Likert scale) Evidence-Based Practice Competency Questionnaire for Professional Registered Nurses (EBP-COQ Prof) was used to assess the competence level of nurses, focusing on attitudes, skills, and knowledge, as well as the utilization of EBP in clinical practice. One-way ANOVA and Pearson coefficient tests were applied to compare the possible differences among variables (two or more groups) as appropriate. A multi-factorial regression model was applied to explore participants' qualifications, including demographics (MSc degree, gender, English language knowledge, etc.) as independent variables, and to control for potential confounding effects toward EBP competency. The $p$-values $< 0.05$ were considered statistically significant. The mean age of the 473 participants (402 women and 71 men) was $44.7 \pm 9.2$ years old. The mean value of competence subscales was found as follows: attitudes $3.9 \pm 0.6$, knowledge $3.7 \pm 0.6$, skills $3.1 \pm 0.8$, and utilization $3.4 \pm 0.7$. A multivariate regression analysis revealed that associates of "Master's degree" (t = 3.039, $p = 0.003$), "Writing an academic article" (3.409, $p = 0.001$), "Working in a University clinic" (2.203, $p = 0.028$), and "Computer Skills" (2.404, $p = 0.017$) positively affected "Attitudes", "Knowledge", "Skills", and "Utilization", respectively. The research data suggest that nurses working in the Greek NHS were limited in competence regarding EBP in comparison with other European countries. Therefore, vocational, educational, and training programs tailored to EBP enhancement are crucially important. This study was not registered.

**Keywords:** evidence-based practice; nurses; competency; attitude; knowledge; skills; utilization

## 1. Introduction

Evidence-based practice (EBP) results in improved healthcare quality and health outcomes, enhances the reliability of healthcare, and reduces variations in care and costs [1–4]. Nurses' competency is defined as "an expected and measurable level of nursing performance that integrates knowledge, skills, abilities, and judgment, based on established scientific knowledge and expectations in nursing practice" [1,5]. For this reason, researchers stress the need for cultivating nurses' EBP competencies, even from the undergraduate level, and focus on designing educational programs and digital tools for undergraduate and postgraduate students, as well as professional nurses [6–13]. In this direction, health organizations aim to reach a high level of practice and the best quality patient outcomes, while they highlight the necessity to evaluate nurses' EBP in order to provide evidence-based, high-quality, and cost-effective care [1]. Healthcare systems and hospitals should provide

the proper equipment and educational support in order to develop EBP competencies in their clinicians and nurses [14].

An increasing number of different scales have been used to evaluate nurses' EBP across Europe. One of these, namely, the EBP-COQ Prof by the Spanish National Survey [15], showed that it could be a beneficial tool when evaluating factors such as levels of education and years of training post-degree, along with active scientific interest, such as reading articles and attending seminars. These factors were considered to enhance and develop EBP competency [16]. However, a systematic review by Saunders et al. [17] stated that EBP competencies alone are not sufficient and cannot be used exclusively in order to reach a high level of EBP in healthcare practice. The same review concluded that competency assessment should be used as an aspect of the development and evaluation of clinical practice and that there is a need for validated tools in order to measure EBP competency that can be more accurate than mere self-assessments [17].

Based on an updated literature review, in a European context, a consensus of experts from the Czech Republic, Greece, Italy, Poland, Slovenia, and Spain identified 24 EBP competencies and 120 learning outcomes for general nurses and advanced practice nurses in order to produce a common tool that can be used in order to assess nursing competency; this can be used for educational intervention and the integration of EBP into daily clinical practice [10].

According to the literature, there seems to be a lack of validated questionnaires in Greek health systems. Major strategies to evaluate nurses' competency are required to focus on training programs and practices in order to improve everyday health practice. There has been an attempt by Patelarou et al. [18] to translate and validate the EBP-COQ questionnaire to evaluate EBP competency in nursing students.

In a recent study conducted by Schetaki et al. [19] in Greece, as precisely defined by Melnyk [2], it was shown that the EBP-COQ Prof questionnaire, which was translated and validated in Greek, is able to measure and evaluate the competency of nurses in the National Health Service in Greece. This was confirmed not only in terms of knowledge, skills, and attitude but also in the utilization of EBP in their daily clinical settings.

The present study aimed (a) to investigate the practice, attitude, knowledge, and skills of Greek nurses regarding EBP, and (b) to determine the association between EBP competency and nurses' demographic characteristics (personal and professional factors).

## 2. Materials and Methods

### 2.1. Study Design and Sample

This study was a cross-sectional survey and utilized a descriptive study design to investigate Greek nurses' practice, attitude, knowledge, and skills regarding EBP and the demographic factors associated with them. A convenience sample was recruited from nurses employed by ten hospitals in Greece (four from the city of Athens and six from Crete Island). This study was conducted from October to December 2020. The sample was recruited from different types of nursing units with the following inclusion criteria: (1) registered nurse as a professional role, (2) age > 18 years, (3) ability to read in Greek, and (4) working experience of more than six months. Six hundred questionnaires were distributed, of which 473 completed questionnaires were returned (a response rate of 78%).

#### 2.1.1. Data Collection Tools

This study used two tools to collect data: the Evidence-Based Practice Competency Questionnaire for Professional Registered Nurses (EBP-COQ Prof) and a demographic survey that was developed specifically for the purposes of the present study.

The EBP-COQ Prof was developed and psychometrically tested to measure Spanish-speaking nurses' attitudes toward evidence-based practice [15]. Schetaki et al. [19] translated the EBP-COQ Prof into Greek, establishing the scale's validity and reliability for Greek nurses (Cronbach $\alpha$: attitudes (8 items, $\alpha = 0.89$), knowledge (11 items, $\alpha = 0.94$), skills (6 items, $\alpha = 0.82$), and utilization (10 items, $\alpha = 0.87$). The EBP-COQ Prof scale consists

of 35 items that attempt to assess nursing staff competence, with a focus on attitudes, skills, and knowledge of EBP and the use of EBP in clinical practice. These 35 items are in the form of a 5-point Likert scale (1 strongly disagree, 2 disagree, 3 neither agree nor disagree, 4 agree, and 5 totally agree), with the higher values corresponding to a higher rating, indicating a positive rating attitude toward EBP.

In order to determine factors that are associated with different attitudes about EBP, we developed a form to gather personal and professional information from each of the subjects (age, biological sex, possession of a master's degree, years of nursing experience, writing academic/professional nursing articles in the last 5 years, level of English language knowledge, level of computer knowledge).

### 2.1.2. Ethical Consideration

The Hellenic Mediterranean University Ethics Committee (no. 28/18.01.21) examined and approved this study. The present study was conducted in accordance with the new General Data Protection Regulation (GDPR) (EU 2016/679) on sensitive personal data, which went into effect on 25 May 2018. The required licenses were obtained by the respective services prior to installation. The data obtained were anonymous, and their usage was limited to the survey and the principal researcher's access to them. The participants gave their written agreement after being properly informed that the procedure was anonymous, that their personal data and replies would be used solely for research reasons, and that they may leave at any moment.

### 2.1.3. Data Analysis

We used SPSS version 25.0 (SPSS Inc., Chicago, IL, USA) for the statistical analysis [20]. Continuous data were reported as mean $\pm$ sd, whereas categorical variables were expressed as absolute numbers (N) and percentages (%). The Shapiro–Wilk test was used to determine the normality of the variables, as well as the "Normal Q-Q plot", "Detrended Normal Q-Q plot", and "Box Plot" studies. Statistical differences were considered to be significant at $p < 0.05$ for all tests. When comparing more than two group means, we used the independent t-test, and when comparing more than two group means, we used one-way ANOVA. The Pearson coefficient was used to connect two continuous variables. The point biseral coefficient was used to correlate a continuous variable with an ordinal variable. For the purposes of the analysis of confounding factors associated with the EBP-COQ Prof subscales, we used a multiple linear regression model. In the univariate analysis, the criterion for the initial entry of variables into multiple regression models was $p < 0.25$.

## 3. Results

### 3.1. Descriptive Analysis

This study sample consisted of 473 participants (402 women and 71 men), with an average age of 44.7 years (sd = 9.2). The average time of nursing experience was 17.3 years (sd = 10.1), and one hundred and seventy-one (36.5%) were working in a university clinic.

One hundred and eighteen nurses (25.0%) held a master's degree and fifty-three (11.2%) had written at least one academic or professional nursing article in the last 5 years. Twenty nurses (4.3%) had no knowledge of English at all and eighty-six (18.3%) had excellent knowledge. Three nurses (0.6%) had no knowledge about computers at all and one hundred and nineteen (25.4%) had excellent knowledge (Table 1).

**Table 1.** Characteristics of sample.

|  | N | % |
|---|---|---|
| Biological sex |  |  |
| Male | 71 | 15.0% |
| Female | 402 | 85.0% |
| Age (mean $\pm$ sd) | 44.7 $\pm$ 9.2 |  |

**Table 1.** *Cont.*

|  | N | % |
|---|---|---|
| Master's degree |  |  |
| No | 354 | 75.0% |
| Yes | 118 | 25.0% |
| Writing academic/professional nursing articles in the last 5 years |  |  |
| No | 420 | 88.8% |
| Yes | 53 | 11.2% |
| English language knowledge |  |  |
| Not at all | 20 | 4.3% |
| Moderate | 108 | 23.0% |
| Good | 255 | 54.4% |
| Excellent | 86 | 18.3% |
| Computer knowledge |  |  |
| Not at all | 3 | 0.6% |
| Moderate | 83 | 17.7% |
| Good | 264 | 56.3% |
| Excellent | 119 | 25.4% |
| Work in a university clinic |  |  |
| No | 297 | 63.5% |
| Yes | 171 | 36.5% |
| Years of nursing experience (mean ± sd) | 17.3 ± 10.1 | |

N: absolute number, %: percentage.

As for the EBP-COQ Prof, the mean of attitudes was 3.9 (sd = 0.6), knowledge was 3.1 (sd = 0.8), skills was 3.7 (sd = 0.6), and utilization was 3.4 (sd = 0.7).

The subscales attitudes (8 items, $\alpha$ = 0.918), knowledge (11 items, $\alpha$ = 0.952), skills (6 items, $\alpha$ = 0.930), and utilization (10 items, $\alpha$ = 0.940) were found to be highly reliable.

### 3.1.1. Correlations of EBP-COQ Prof Subscales
Bivariate Analysis

Table 2 summarizes the results of the correlations of the subscales of the EBP-COQ Prof questionnaire with the studied factors. According to the results, older ages corresponded to higher values of the skills (r = 0.153, $p$ = 0.001) and utilization (r = 0.132, $p$ = 0.004) subscales. The levels of attitudes (t = −4.86, $p$ < 0.001), knowledge (t = −9.17, $p$ < 0.001), skills (t = −5.29, $p$ < 0.001), and utilization (t = −3.29, $p$ < 0.001) were statistically significantly higher for nurses who had a master's degree than nurses who did not. Moreover, the levels of attitudes (t = −3.87, $p$ < 0.001), knowledge (t = −6.50, $p$ < 0.001), and skills (t = −4.00, $p$ < 0.001) were statistically significantly higher for nurses who had written at least one academic or professional nursing article in the last 5 years than nurses who had not. A higher level of computer knowledge corresponded to higher values of the attitude (rpb = 0.138, $p$ = 0.003), knowledge (rpb = 0.290, $p$ < 0.001), skills (rpb = 0.161, $p$ < 0.001), and utilization (rpb = 0.113, $p$ = 0.015) subscales. More years of nursing experience corresponded to higher values of the skills (r = 0.179, $p$ < 0.001) and utilization (r = 0.140, $p$ = 0.002) subscales. The levels of attitudes (t = −3.25, $p$ = 0.001), knowledge (t = −3.72, $p$ < 0.001), skills (t = −2.88, $p$ = 0.004), and utilization (t = −4.32, $p$ < 0.001) were statistically significantly higher for nurses who worked in a university clinic. English language knowledge was significantly higher for those with higher levels of the attitude (rpb = 469, $p$ = 0.012), knowledge (rpb = 469, $p$ < 0.001), and skills (rpb = 469, $p$ = 0.012) subscales. The findings revealed no significant correlation between biological sex and the EBP-COQ Prof subscales.

**Table 2.** Correlations of the subscales of the EBP-COQ Prof questionnaire with the studied factors.

|  | Attitudes | Knowledge | Skills | Utilization |
|---|---|---|---|---|
| **Age** | r(473) = −0.026, $p = 0.814$ | r(473) = −0.011, $p = 0.814$ | r(473) = 0.153, $p = 0.001$ | r(473) = 0.132, $p = 0.004$ |
| **Biological sex** | t(471) = −1.448, $p = 0.148$ | t(471) = 1.857, $p = 0.064$ | t(471) = 0.045, $p = 0.964$ | t(471) = −0.616, $p = 0.538$ |
| **Possession of a master's degree** | t(470) = −4.862, $p < 0.001$ | t(470) = −9.169, $p < 0.001$ | t(470) = −5.289, $p < 0.001$ | t(470) = −3.294, $p < 0.001$ |
| **Writing academic/professional nursing articles in the last 5 years** | t(471) = −3.870, $p < 0.001$ | t(471) = −6.496, $p < 0.001$ | t(471) = −4.003, $p < 0.001$ | t(471) = −1.939, $p = 0.053$ |
| **Level of English language knowledge** | rpb(469) = 0.116, $p = 0.012$ | rpb(469) = 0.257, $p < 0.001$ | rpb(469) = 0.116, $p = 0.012$ | rpb(469) = 0.064, $p = 0.169$ |
| **Level of computer knowledge** | rpb(469) = 0.138, $p = 0.003$ | rpb(469) = 0.290, $p < 0.001$ | rpb(469) = 0.161, $p < 0.001$ | rpb(469) = 0.113, $p = 0.015$ |
| **Years of nursing experience** | r(466) = 0.010, $p = 0.822$ | r(466) = −0.003, $p = 0.951$ | r(466) = 0.179, $p < 0.001$ | r(466) = 0.140, $p = 0.002$ |
| **Work in a university clinic** | t(466) = −3.253, $p = 0.001$ | t(466) = −3.718, $p < 0.001$ | t(466) = −2.883, $p = 0.004$ | t(466) = −4.324, $p < 0.001$ |

r: Pearson coefficient, rpb: point biseral coefficient, t: Student's t-test, *p*: *p*-value.

Multivariable Analysis

The multiple regression model statistically significantly predicted the subscale attitudes (F(6, 457) = 7.163, $p < 0.001$, adj. $R^2 = 0.074$). The variables "Master's degree" ($p = 0.003$), "Work in a university clinic" ($p = 0.008$), and "Writing academic/professional nursing articles in the last 5 years" ($p = 0.024$) statistically significantly added to the prediction. The regression coefficients and standard errors are presented in Table 3.

**Table 3.** Multiple regression results for attitudes.

| Model Dependent Variable—Attitudes | Unstandardized Coefficients | | t | Sig. | 95.0% Confidence Interval for B | |
|---|---|---|---|---|---|---|
|  | B | Std. Error | | | Lower Bound | Upper Bound |
| **(Constant)** | 3.322 | 0.178 | 18.642 | 0.000 | 2.972 | 3.672 |
| **Biological sex** | 0.148 | 0.078 | 1.905 | 0.057 | −0.005 | 0.300 |
| **Master's degree** | 0.210 | 0.069 | 3.039 | 0.003 | 0.074 | 0.347 |
| **Writing academic/professional nursing articles in the last 5 years** | 0.211 | 0.093 | 2.271 | 0.024 | 0.028 | 0.393 |
| **English language knowledge** | 0.007 | 0.046 | 0.151 | 0.880 | −0.083 | 0.097 |
| **Computer knowledge** | 0.075 | 0.051 | 1.469 | 0.142 | −0.025 | 0.175 |
| **Work in a university clinic** | 0.153 | 0.058 | 2.647 | 0.008 | 0.039 | 0.266 |

The multiple regression model statistically significantly predicted the subscale knowledge (F(6, 457) = 23.306, $p < 0.001$, adj. $R^2 = 0.224$). The variables "Master's degree" ($p < 0.001$), "Writing academic/professional nursing articles in the last 5 years" ($p = 0.001$), "Computer knowledge" ($p = 0.002$), and "Work in a university clinic" ($p = 0.006$) statistically significantly added to the prediction. The regression coefficients and standard errors are presented in detail in Table 4.

**Table 4.** Multiple regression results for knowledge.

| Model Dependent Variable—Knowledge | Unstandardized Coefficients | | t | Sig. | 95.0% Confidence Interval for B | |
|---|---|---|---|---|---|---|
| | B | Std. Error | | | Lower Bound | Upper Bound |
| (Constant) | 2.551 | 0.223 | 11.436 | 0.000 | 2.112 | 2.989 |
| Biological sex | −0.102 | 0.097 | −1.055 | 0.292 | −0.293 | 0.088 |
| Master's degree | 0.533 | 0.087 | 6.148 | 0.000 | 0.363 | 0.703 |
| Writing academic/professional nursing articles in the last 5 years | 0.396 | 0.116 | 3.409 | 0.001 | 0.168 | 0.625 |
| English language knowledge | 0.067 | 0.057 | 1.164 | 0.245 | −0.046 | 0.180 |
| Computer knowledge | 0.195 | 0.064 | 3.058 | 0.002 | 0.070 | 0.321 |
| Work in a university clinic | 0.201 | 0.072 | 2.789 | 0.006 | 0.059 | 0.343 |

The multiple regression model statistically significantly predicted the subscale skills $(F_{(7, 449)} = 9.598, p < 0.001, \text{adj. } R^2 = 0.117)$. The variables "Computer knowledge" $(p = 0.002)$, "Master's degree" $(p = 0.011)$, "Writing academic/professional nursing articles in the last 5 years" $(p = 0.019)$, and "Work in a university clinic" $(p = 0.028)$ statistically significantly added to the prediction. Table 5 presents the regression coefficients and standard errors.

**Table 5.** Multiple regression results for skills.

| Model Dependent Variable—Skills | Unstandardized Coefficients | | t | Sig. | 95.0% Confidence Interval for B | |
|---|---|---|---|---|---|---|
| | B | Std. Error | | | Lower Bound | Upper Bound |
| (Constant) | 2.820 | 0.253 | 11.124 | 0.000 | 2.322 | 3.318 |
| Age | 0.004 | 0.007 | 0.564 | 0.573 | −0.009 | 0.016 |
| Master's degree | 0.185 | 0.073 | 2.541 | 0.011 | 0.042 | 0.329 |
| Writing academic/professional nursing articles in the last 5 years | 0.226 | 0.096 | 2.351 | 0.019 | 0.037 | 0.415 |
| English language knowledge | 0.005 | 0.048 | 0.100 | 0.921 | −0.089 | 0.098 |
| Computer knowledge | 0.169 | 0.055 | 3.098 | 0.002 | 0.062 | 0.276 |
| Years of nursing experience | 0.011 | 0.006 | 1.898 | 0.058 | 0.000 | 0.023 |
| Work in a university clinic | 0.133 | 0.060 | 2.203 | 0.028 | 0.014 | 0.251 |

The multiple regression model statistically significantly predicted the subscale utilization $(F_{(7, 449)} = 6.299, p < 0.001, \text{adj. } R^2 = 0.075)$. The variables "Work in a university clinic" $(p < 0.001)$ and "Computer knowledge" $(p = 0.017)$ statistically significantly added to the prediction. The regression coefficients and standard errors can be found in Table 6.

**Table 6.** Multiple regression results for utilization.

| Model Dependent Variable—Utilization | Unstandardized Coefficients | | t | Sig. | 95.0% Confidence Interval for B | |
|---|---|---|---|---|---|---|
| | B | Std. Error | | | Lower Bound | Upper Bound |
| (Constant) | 2.457 | 0.265 | 9.264 | 0.000 | 1.936 | 2.978 |
| Age | 0.011 | 0.007 | 1.544 | 0.123 | −0.003 | 0.024 |
| Master's degree | 0.113 | 0.076 | 1.487 | 0.138 | −0.036 | 0.263 |
| Writing academic/professional nursing articles in the last 5 years | 0.093 | 0.101 | 0.922 | 0.357 | −0.105 | 0.290 |

**Table 6.** *Cont.*

| Model Dependent Variable—Utilization | Unstandardized Coefficients | | t | Sig. | 95.0% Confidence Interval for B | |
|---|---|---|---|---|---|---|
| | B | Std. Error | | | Lower Bound | Upper Bound |
| English language knowledge | −0.010 | 0.050 | −0.207 | 0.836 | −0.108 | 0.088 |
| Computer knowledge | 0.137 | 0.057 | 2.404 | 0.017 | 0.025 | 0.249 |
| Years of nursing experience | 0.003 | 0.006 | 0.407 | 0.684 | −0.010 | 0.015 |
| Work in a university clinic | 0.243 | 0.063 | 3.856 | 0.000 | 0.119 | 0.367 |

## 4. Discussion

In this study, the Greek version of the EBP-COQ Prof questionnaire was used, which consists of 35 statements and is a reliable tool [19], to investigate the competency of the nurses of the national health system in Greece (NHS) in terms of attitude, skills, knowledge, and utilization regarding EBP in daily clinical practice. As a reliable tool, it helped to reveal the state of competency toward EBP of registered nurses in the NHS in Greece and might give the opportunity to improve the competency with suitable interventions. This study was conducted using a convenience sample of 473 nurses working in 10 hospitals. The mean age of 473 participants (402 women and 71 men) was $44.7 \pm 9.2$ years old. Our results show that the nursing staff presented a positive attitude with a mean of 3.9 (sd = 0.6) and skills of 3.7 (sd = 0.6), and presented lower values in utilization with 3.4 (sd = 0.7) and knowledge with 3.1 (sd = 0.8). The factors that influenced these variables were "Master's degree" ($t = 3.039$, $p = 0.003$), "Writing an academic article" (3.409, $p = 0.001$), "Working in a University clinic" (2.203, $p = 0.028$), and "Computer Skills" (2.404, $p = 0.017$).

However, in research conducted in Spain by Ramos et al. [16] using the Spanish version of the same questionnaire [15], the majority of nurses showed a higher mean for attitude, followed by skills, and lower for the dimensions of knowledge and utilization. These findings are similar to our study, but there is a difference in the dimension of knowledge which is higher than utilization [16]. Perhaps this was due to the high percentage of participants with master's and doctoral degrees in the Spanish study. According to the results of other research, it seems that nurses worldwide lack EBP readiness. Although there is a positive attitude and recognition of the significance of the value of EBP [21–23], on the other hand, there is a lack of competency, knowledge, and skills for the implementation of EBP [7,21,24–26]. In China, in 2019, a survey of manager nurses reported a positive attitude toward EBP but rarely implemented it due to a lack of knowledge [27]. According to Saunders et al. [28], due to this lack of knowledge, participants were not able to recognize when EBP was being used in clinical practice. Another common finding of many studies, as in the present one, is the low implementation of EBP [28–34]. It seems that some of the barriers that can affect the implementation are the lack of time, an unsupportive working environment, the daily routine, and the difficulty of having access to information resources [35]. Unexpectedly, the research findings by Aynalem et al. [29] showed that 48.9% of the participants had an unfavorable attitude toward EBP, and 21.6% stated that they did not wish to use EBP in their clinical practice. A negative attitude was also seen in other studies [36,37].

The present study investigated the variables related to attitude, knowledge, skills, and utilization in terms of EBP. Regarding age, it seems to have no significant correlation with our subscales. In contrast, in a previous study, it was stated that participants younger than 30 years old had a significant correlation with the competency of EBP ($t = 2.163$, $p < 0.05$) [38].

In our study, a factor that significantly affected the subscales was the level of education. Based on the findings, the higher the educational level, the higher the subscale score. Therefore, nurses who possessed master's or doctorate degrees had a positive attitude and better knowledge and skills than those who possessed a bachelor's degree. This was also confirmed in previous studies in the literature [7,16,23,26,37,39]. Nevertheless, a study

conducted in Saudi Arabia surprisingly found that the level of education did not have a positive influence in terms of attitude, knowledge, and utilization [40].

This study revealed that nursing staff with a higher level of computer skills corresponds to higher values of the knowledge, skills, and utilization subscales. The results of the present study share several similarities with Patelarou et al.'s [18] study, which showed that having good computer skills was found to positively influence the nursing students' perceptions of EBP, where the median value in the VAS scale was 7.0. In addition to other research in the literature, it was found that the use and accessibility of a computer significantly affected the utilization of EBP [29,41].

Our survey analysis found no correlation between the years of nursing experience and the subscales; Tomotaki et al. [7] stated that the years of working experience are not necessary to have a correlation with EBP competency. Farokhzadian et al. [36] support the idea that the working experience helps to be confident in using EBP in daily clinical practice. However, this study has not confirmed previous research that stated that nursing staff with fewer years of professional experience had better utilization of EBP [29]. Remarkably, a negative correlation was reported in the survey conducted by Heydari et al. [37], revealing that working experience negatively affects the knowledge, skills, and utilization regarding EBP.

Another parameter that is related to and affected our variables were the levels of attitude, knowledge, skills, and utilization, which were statically significantly higher for nurses who worked in a university clinic than nurses not working in a clinic. This concurs well with previous findings in the literature that show a significant difference depending on the type of hospital and wards where the nurses work [26,37,38].

According to our study results, the attitude, knowledge, and skills levels were statistically higher for nurses who had written at least one academic or professional nursing article in the last five years than those who had not. The results of this study are consistent with previous studies that stated that there was a positive correlation with competency in EBP. Spanish researchers found that reading scientific articles increased competency in EBP [16]. Tomotaki et al. [7] found that those conducting research are positively related to attitude, knowledge, and skills regarding EBP. Moreover, Yoo et al. [39] showed that involvement in the research positively affects the utilization of EBP. In addition, the research of Alqahtani et al. [40] showed a positive correlation between knowledge and conducting research.

Our results show that the English language level had no significant correlation with attitude, knowledge, skills, and utilization. Regarding this point, Patelarou et al. [18] revealed in their study that the higher the knowledge of the English language, the lower their perception of EBP (median = 8.0 on the VAS scale).

The analysis did not identify any significant biological sex differences in the variables attitude, knowledge, skills, and utilization. These findings are in complete agreement with the study conducted by Heydari et al. [37]. The most surprising is that the survey conducted by Patelarou et al. [18] stated that men had higher mean values regarding knowledge and skills in EBP ($28.2 \pm 4.3$) compared with women ($26.1 \pm 5.3$).

## 5. Strengths and Limitations of This Study

It is noteworthy that no other scale has been used in the Greek language that can investigate and assess the competence of registered nurses in the NHS regarding EBP. This is the first study revealing results about Greek nurses' attitudes, skills, knowledge, and utilization. The questionnaire is self-administered, and therefore, there should be a degree of caution as to the validity of the responses, but a sufficiently large sample and convenience sampling tend to eliminate this disadvantage. During the period that this survey was conducted, there was a lockdown due to the COVID-19 pandemic and the accessibility to the hospitals was not free.

## 6. Conclusions

This study revealed that Greek NHS nurses had a positive attitude, moderate levels of skills, and lower levels of utilization and knowledge. The factors influencing these variables were educational level, writing an academic article in the last five years, level of computer knowledge, and the type of clinic they worked in. The limited literature in the Greek area around the specific subject confirms the requirement for further research in order to identify the needs. For this reason, educational programs should be developed that will aim at continuous training to enhance and improve the appropriate culture and guarantee a favorable climate toward EBP.

**Author Contributions:** Conceptualization, S.S. and A.P.; methodology, A.P.; formal analysis, S.S. and A.P.; investigation, data curation, S.S., K.G. and A.P.; writing—original draft preparation, S.S., E.P. and A.P.; writing—review and editing, S.S., E.P., K.G., C.K. and A.P.; supervision, A.P. All authors have read and agreed to the published version of the manuscript.

**Funding:** This research received no external funding.

**Institutional Review Board Statement:** This research was approved by the Hellenic Mediterranean University Ethics Committee. Respondents were informed by an information sheet about the purpose of this research, asking them to give their full consent for participation. The whole research process respected the dignity of the participants, protected their privacy and anonymity, and ensured an adequate level of confidentiality. The data were used only for the purpose of the present study.

**Informed Consent Statement:** Informed consent was obtained from all subjects involved in this study.

**Data Availability Statement:** The data presented in this study are available on request from the corresponding author.

**Public Involvement Statement:** No public involvement in any aspect of this research.

**Guidelines and Standards Statement:** This manuscript was drafted against the STROBE (Strengthening the Reporting of Observational Studies in Epidemiology) statement for observational studies (cross-sectional).

**Conflicts of Interest:** The authors declare no conflict of interest.

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
