# Peer review of "Evidence-Based Practice Competency of Registered Nurses in the Greek National Health Service"

_nursrep, doi:10.3390/nursrep13030105_

Round 1

Reviewer 1 Report

The research data suggest that nurses working in Greek NHS are limited in competence towards EBP in comparison to other European countriesï¼› I consider this study relevant in the field  of EBP of Greek NHSï¼›This study adds greek data if compared with other published material; The references are appropriate.

In my opionion the manuscript is clear, relevant for the field and presented in a well-structured manner.

The figures/tables/images/schemes are appropriate.

The limited literature in the Greek area around the specific subject confirms the requirement for further research in order to identify the needs. For this reason, educational programs should be developed to fill the gap in EBP competence of registred nurses.

The conclusions consistent with the evidence and arguments presented.

The English language is appropriate 

Author Response

We would like to warmly thank the reviewers for their kind words and valuable suggestions, which have significantly contributed to improving this manuscript. The manuscript has been revised with track changes.

Response to Reviewer 1 Comments

Comments and Suggestions for Authors

The research data suggest that nurses working in Greek NHS are limited in competence towards EBP in comparison to other European countriesï¼› I consider this study relevant in the field  of EBP of Greek NHSï¼›This study adds greek data if compared with other published material; The references are appropriate.

In my opionion the manuscript is clear, relevant for the field and presented in a well-structured manner.

The figures/tables/images/schemes are appropriate.

The limited literature in the Greek area around the specific subject confirms the requirement for further research in order to identify the needs. For this reason, educational programs should be developed to fill the gap in EBP competence of registred nurses.

The conclusions consistent with the evidence and arguments presented

The English language is appropriate 

Response: Thank you very much for your valuable comments. Your comments encourage us and give us motivation to continue the research in the field.

Reviewer 2 Report

The manuscript “Evidence-Based Practice Competency of registered nurses in the Greek National Health Service” merit and represents a major step towards understanding evidence-based practice competence of registered nurses in Greece, there are several limitations and shortcomings which need to be addressed before acceptance:

1) In line 28, does the word “Respectively.” belong to the previous or following sentences?

2) In line 37, 73, 187 and 214, please remove the superfluous spaces.

3) Please add the Reference doi 10.3390/jpm11070623, when you describe the educational programs in line 41.

4) Table 1 should be formatted more clearly, in which the questions are left-justified.

5) A paragraph should be inserted between Table 1 and the following text, otherwise the reader cannot be sure that it is a continuation of the text or a supplement to the table. Same after Table 3 and Table 4.

6) The heading of figure 1 should not be written in bold.

7) You should decide whether to put spaces to achieve consistent formatting. For example, whether you want to write p=0, (like in the abstract) or p = 0, (like for example in Table 2 and even other parts in your manuscript).

8) Sometimes SD is written in capital letters, sometimes not. Please decide on a formatting style.

9) You should explain the abbreviations you use (like in table 2 the letters t, r, rpb). You should doublecheck the labeling of all figures and tables.

10) In line 226, you should give possible reasons for this, such as the working conditions in shift work and the lack of time resources of mostly female middle-aged nurses due to afterwork family care work.

11) The limitations of the study should be further clarified regarding the composition of the study participants.

I look forward to your reply and wish you success for the resubmission.

Author Response

We would like to warmly thank the reviewers for their kind words and valuable suggestions, which have significantly contributed to improving this manuscript. The manuscript has been revised with track changes.

Response to Reviewer 2 Comments

Comments and Suggestions for Authors

Reviewer 2

The manuscript “Evidence-Based Practice Competency of registered nurses in the Greek National Health Service” merit and represents a major step towards understanding evidence-based practice competence of registered nurses in Greece, there are several limitations and shortcomings which need to be addressed before acceptance:

Point 1: In line 28, does the word “Respectively.” belong to the previous or following sentences?

Response: Many thanks for your comment, the word “Respectively” has been corrected.

Point 2: ) In line 37, 73 , 187 and 214, please remove the superfluous spaces

Response: Thank you very much, we have removed the superfluous spaces.

Point 3:  Please add the Reference doi 10.3390/jpm11070623, when you describe the educational programs in line 41p

Response: Thank you for your precious advice, we have added this Reference.

Point 4: Table 1 should be formatted more clearly, in which the questions are left-justified.

Response: Many thanks for your comment. We have formatted Table 1 in order to be clearer.

Point 5: A paragraph should be inserted between Table 1 and the following text, otherwise the reader cannot be sure that it is a continuation of the text or a supplement to the table. Same after Table 3 and Table 4

Response: Thank you very much, we have inserted a paragraph between the Tables and the following texts.

Point 6: The heading of figure 1 should not be written in bold

Response: Many thanks for the comment but the figure has been deleted as suggested by a reviewer.

Point 7: You should decide whether to put spaces to achieve consistent formatting. For example, whether you want to write p=0, (like in the abstract) or p = 0, (like for example in Table 2 and even other parts in your manuscript)

Response: Thank you for your valuable comment, it has been corrected with the same format in the whole manuscript.

Point 8: Sometimes SD is written in capital letters, sometimes not. Please decide on a formatting style

Response: Many thanks, the text has been revised accordingly.

Point 9: You should explain the abbreviations you use (like in table 2 the letters t, r, rpb). You should doublecheck the labeling of all figures and tables

Response: Thank you for your precious comment, we have explained every abbreviation and the labeling of figures and tables has been checked.

Point 10: In line 226, you should give possible reasons for this, such as the working conditions in shift work and the lack of time resources of mostly female middle-aged nurses due to afterwork family care work

Response: Thank you for your valuable comments. You have given us the opportunity to explain some of the barriers that nurses may face during the EBP implementation process.

Point 11: The limitations of the study should be further clarified regarding the composition of the study participants

Response: Many thanks to the reviewer for this valuable comment, which has given us the opportunity to add more information regarding the limitations of the survey.

We are grateful to you for your important contribution to improving the manuscript.

Reviewer 3 Report

Dear authors,

to assess Nurses' competency towards Evidence-based Practice is very important to enhance the knowledge of this competence in the international context. After adapting and validating the Greek version it is important to conduct the study on a large sample of nurses in Greece. Congratulations.

Keywords: "Competency" is not a MeSH term (I suggest the MeSH: "Clinical Competence"); "Skills" and "Utilization" are not a MeSH terms (you should use another term for correct indexing).

I have some comments to enhance the manuscript according STROBE statement.

Aim should be described at the end of the introduction.

Materials and methods: In the subheading study design and sample, exclusion criteria would need to be indicated (e.g. years of experience); study size should be stimated to explain how the study size was arrived at (in my opinion, the proportion of this sample estimatation can be made on the basis of the previous study's standard deviation).

The date on which the instrument was applied is added in the abstract, but not added in the manuscript text.

Results: In my opinion, figure 1 is unnecessary. It does not provide any additional information to the text.

Acronyms used should be explained at the bottom of the tables.

Unify criteria for describing decimal places in the whole text and in tables (0.000 vs .000).

In the discussion, page 8 (line 220), the citation to Ramos et al (2021) seems to use APA style. Idem to: Aynalem et al (line 234), Patelarou et al (line 251), tomotaki (line 257). Please, review all the discussión in this sense. Review this also in the introduction.

There is a lack of discussion of the results of this study with the results of the initial validation study of the instrument in Greece.

Author Response

We would like to warmly thank the reviewers for their kind words and valuable suggestions, which have significantly contributed to improving this manuscript. The manuscript has been revised with track changes.

Response to Reviewer 3 Comments

Reviewer 3

Dear authors,

to assess Nurses' competency towards Evidence-based Practice is very important to enhance the knowledge of this competence in the international context. After adapting and validating the Greek version it is important to conduct the study on a large sample of nurses in Greece. Congratulations.

Point 1: Keywords: "Competency" is not a MeSH term (I suggest the MeSH: "Clinical Competence"); "Skills" and "Utilization" are not a MeSH terms (you should use another term for correct indexing).

Response: Thank you for your valuable comment. We would be thankful if you accepted these keywords as vital components of the research tool used in the present study and moreover as the most popular in the scientific literature in the field of EBP.

Point 2: Materials and methods: In the subheading study design and sample, exclusion criteria would need to be indicated (e.g. years of experience); study size should be stimated to explain how the study size was arrived at (in my opinion, the proportion of this sample estimatation can be made on the basis of the previous study's standard deviation).

Response: Thank you for your precious comment, we have added more information about the criteria and the study size.

Point 3: Results: In my opinion, figure 1 is unnecessary. It does not provide any additional information to the text

Response: Thank you for your valuable comments, the figure 1 has been deleted as you suggest.

Point 4: Acronyms used should be explained at the bottom of the tables

Response: Many thanks, we have given an explanation in every acronym of the tables.

Point 5: Unify criteria for describing decimal places in the whole text and in tables (0.000 vs .000)

Response: Thank you very much for your valuable comments, we have fixed the description of the decimal places in the whole text and the tables.

Point 6: In the discussion, page 8 (line 220), the citation to Ramos et al (2021) seems to use APA style. Idem to: Aynalem et al (line 234), Patelarou et al (line 251), tomotaki (line 257). Please, review all the discussión in this sense. Review this also in the introduction.

Response: Thank you for your precious comment. We have revised the manuscript in ACS style as recommended by the Journal.

Point 7: There is a lack of discussion of the results of this study with the results of the initial validation study of the instrument in Greece

Response: Thank you for the opportunity you gave us to add more information about the instrument we used.

We feel grateful for your valuable comment that you have given us the opportunity to improve our manuscript.